EMBO
Molecular Medicine

# Metabo-epigenetic circuits of heart failure: chromatin-modifying enzymes as determinants of metabolic plasticity

Mark E Pepin [ID] [1,2,3,4,5,10], Xuemin Gong [ID] [1,2,3,10], Almut Schulze[6] & Johannes Backs [ID] [1,2,3,7,8,9 ✉]

## Abstract

**Metabolic adaptations are a functional requirement for the heart to accommodate its broad range of physiologic operating conditions. It is increasingly recognized that persistent and exaggerated metabolic alterations precede adverse cardiac remodeling leading to heart failure. These metabolic shifts are coupled with changes in cardiac gene expression, driven in part by chromatin-modifying enzymes, which have recently been identified as both sensors and transducers of metabolic stress and gene regulatory networks, respectively. This review synthesizes the latest evidence implicating chromatin-modifying enzymes as key regulators of metabolic reprogramming in the heart, providing a framework to understand how metabolic stressors are incorporated as epigenetic modifications that regulate cardiac gene expression. We propose a model of 'metabo-epigenetic circuitry' within which energy metabolic perturbations drive transcriptional and epigenetic changes that ultimately contribute to cardiac dysfunction. Although many nodes in these circuits remain unidentified, this viewpoint opens new avenues for investigating chromatin-modifying enzymes as therapeutic targets to halt the metabolic programs that promote heart failure.**

**Keywords** Metabolism; Epigenetics; Heart Failure; Cardiomyopathy; Signaling
**Subject Categories** Cardiovascular System; Chromatin, Transcription & Genomics; Metabolism

## Introduction

Compared to other tissues, the heart consumes a remarkable quantity of fuel, daily consuming over 30 kg of ATP to meet the body's resting hemodynamic requirements (Dorn, 2013). And yet, cardiomyocytes contain only enough biochemical energy to execute roughly 10 ventricular contractions, relying on a continual influx of circulating metabolic fuels. Maintaining this dynamic equilibrium is further complicated by the heart's wide range of operating conditions, alterations in myocardial substrate delivery, and preferential utilization of individual substrates (Costantino et al, 2019; Lopaschuk and Kelly, 2008). The regulation of cardiac metabolism therefore depends on coupling fuel consumption to the body's ever-changing hemodynamic requirements. Pathological cardiac metabolic shifts are found in an array of clinical contexts, including familial monogenic cardiomyopathies and lifestyle-associated cardiometabolic disease (Aminian et al, 2019; Shah et al, 2020). Familial-based sequencing and genome-wide association studies have together identified over a thousand rare genetic variants that individually correlate with heart disease risk (Burke et al, 2016). However, the prognostic and therapeutic value of these genetic discoveries remains limited owing to their rarity, small effect size, variable penetrance, and unpredictable manifestations— or pleiotropy—even among monogenic etiologies (Mazzarotto et al, 2020). Heart disease risk is also highly variable in response to the most well-studied environmental and lifestyle-associated factors, such as hypertension (Rodeheffer, 2011), ischemic heart disease (Wolk et al, 1972), obesity (Alpert et al, 2014), and diabetes mellitus (Kenny and Abel, 2019). Recent studies have therefore begun to explore the molecular basis of gene-environment interactions (Smith et al, 2010; Pepin et al, 2025), or epigenetic processes, as the key determinants of heart disease susceptibility and pathogenesis.

Although initially discovered for their role in determining cellular fate by regulating developmental programs (Waddington, 2011), epigenomic alterations are now known to encompass adult-onset diseases, and the epigenome remains responsive to both physiologic (Etchegaray and Mostoslavsky, 2016) and pathologic (Joehanes et al, 2016) stimuli. Epigenetic modifications are broadly defined as covalent modifications that occur either directly onto DNA (e.g., cytosine C-5 methylation) or to its auxiliary transcriptional structures (e.g., histone proteins) (Fig. 1). Often, post-translational regulation of mRNA stability

[1]Heidelberg University, Medical Faculty Heidelberg, Institute of Experimental Cardiology, 69115 Heidelberg, Germany. [2]Heidelberg University Hospital, Department of Internal Medicine VIII, 69115 Heidelberg, Germany. [3]German Center for Cardiovascular Research (DZHK), Partner Site Heidelberg/Mannheim, 69120 Heidelberg, Germany. [4]Stanford Cardiovascular Institute, Stanford University School of Medicine, Stanford, CA, USA. [5]Department of Medicine, Division of Cardiovascular Medicine, Stanford University School of Medicine, Stanford, CA, USA. [6]Division of Tumor Metabolism and Microenvironment, German Cancer Research Center (DKFZ) and FSPA-ZMBH Alliance, Heidelberg, Germany. [7]Molecular Medicine Partnership Unit, European Molecular Biology Laboratory (EMBL), 69117 Heidelberg, Germany. [8]Heidelberg University, 69120 Heidelberg, Germany. [9]Helmholtz Institute for Translational AngioCardioScience (HI-TAC) of the Max Delbrück Center for Molecular Medicine in the Helmholtz Association (MDC) at Heidelberg University, Heidelberg 69117, Germany. [10]These authors contributed equally: Mark E Pepin, Xuemin Gong. ✉E-mail: johannes.backs@cardioscience.uni-heidelberg.de

**Glossary**

| Term | Definition |
|---|---|
| **ABHD5** | Cofactor for PNPLA2/ATGL-mediated triglyceride lipolysis; regulates lipid droplet turnover and releases HDAC4-NT to influence nuclear gene programs. |
| **Acetyl-CoA** | Central metabolic intermediate derived from carbohydrates, fatty acids, or acetate; donor substrate for histone acetylation linking metabolism to chromatin regulation. |
| **AMPK (AMP-activated protein kinase)** | Energy-sensing kinase that phosphorylates metabolic enzymes and transcriptional regulators to restore ATP homeostasis. |
| **ATGL (PNPLA2)** | Rate-limiting triglyceride lipase catalyzing the first step of lipolysis; mutations cause neutral lipid storage disease with cardiomyopathy. |
| **BET proteins (e.g., BRD4)** | Bromodomain-containing chromatin readers that bind acetylated histones to regulate transcription; pharmacologic inhibition alters maladaptive cardiac gene programs. |
| **BDH1/OXCT1** | Ketolytic enzymes catalyzing β-hydroxybutyrate oxidation and succinyl-CoA transfer, respectively; essential for cardiac ketone utilization. |
| **CAMK2** | $Ca^{2+}$/calmodulin-dependent protein kinase implicated in excitation–contraction coupling and maladaptive hypertrophic signaling. |
| **Chromatin-modifying enzymes** | Enzymes catalyzing post-translational modifications of histones or covalent DNA modifications that alter chromatin accessibility and transcriptional activity. |
| **DNMT3A** | De novo DNA methyltransferase establishing CpG methylation patterns; regulates metabolic gene expression in cardiomyocytes. |
| **D-2-hydroxyglutarate (D-2-HG)** | Oncometabolite generated by mutant IDH1/2; competitive inhibitor of TET dioxygenases and Jumonji histone demethylases. |
| **EZH2/PRC2** | Histone methyltransferase catalyzing H3K27 trimethylation; represses gene expression and has been implicated in ischemic cardiomyopathy. |
| **Fatty acid oxidation (FAO)** | Mitochondrial β-oxidation pathway generating acetyl-CoA and NADH; predominant energy source in adult myocardium. |
| **GLP-1 receptor agonists (GLP-1RA)** | Incretin mimetics with cardioprotective effects; epigenetic influence in non-cardiac tissues includes modulation of promoter methylation. |
| **HDACs (histone deacetylases)** | Enzyme family removing acetyl groups from histones and non-histone proteins; regulate cardiac gene expression, metabolic substrate preference, and cytoskeletal stability. |
| **HDAC4-NT** | Proteolytic N-terminal fragment of HDAC4 that represses MEF2 activity and prevents maladaptive glycolytic reprogramming. |
| **HFrEF/HFpEF** | Clinical heart failure subtypes characterized by reduced vs preserved left ventricular ejection fraction, respectively. |
| **Hexosamine biosynthetic pathway (HBP)** | Glucose-derived metabolic branch producing UDP-GlcNAc for O-GlcNAc protein modification. |
| **Histones** | Core nucleosomal proteins subject to post-translational modifications that regulate chromatin structure and gene activity. |
| **IDH1/IDH2** | Isocitrate dehydrogenases producing α-ketoglutarate; neomorphic mutations generate D-2-HG and disrupt epigenetic regulation. |
| **Ketone bodies** | Alternative mitochondrial substrates (β-hydroxybutyrate, acetoacetate) oxidized by the heart during stress or nutrient deprivation. |
| **MEF2** | Myocyte enhancer factor 2; transcription factor regulating muscle gene expression and metabolic programs. |
| **$NAD^+$** | Redox cofactor required for sirtuin deacetylases and metabolic enzyme activity; levels influence mitochondrial function and chromatin regulation. |
| **NHE-1 ($Na^+/H^+$ exchanger 1)** | Membrane antiporter modulating intracellular pH and $Ca^{2+}$ handling; indirectly inhibited by SGLT2 inhibitors. |
| **NR4A1** | Nuclear receptor transcription factor regulating glycolytic and HBP genes downstream of MEF2 signaling. |
| **O-GlcNAcylation** | Reversible attachment of N-acetylglucosamine to serine/threonine residues of proteins; integrates nutrient sensing with transcriptional and signaling pathways. |
| **OGT/OGA** | Enzymes catalyzing the addition (OGT) and removal (OGA) of O-GlcNAc modifications. |
| **Oncometabolites** | Metabolic intermediates (e.g., succinate, fumarate, D-2-HG) that inhibit α-ketoglutarate–dependent dioxygenases and alter epigenetic landscapes. |
| **PPARα/PPARγ** | Nuclear hormone receptors regulating fatty acid oxidation (α) and lipid uptake/storage (γ); central regulators of cardiac substrate preference. |
| **SGLT2 inhibitors** | Antidiabetic agents that improve cardiovascular outcomes; mechanisms include modulation of HDACs and SIRT1 in cardiac tissue. |
| **TET enzymes** | Ten-eleven translocation dioxygenases catalyzing DNA demethylation via 5-methylcytosine oxidation. |
| **UDP-GlcNAc** | End product of the HBP; donor substrate for O-GlcNAcylation of nuclear and cytosolic proteins. |
| **Warburg effect** | Aerobic glycolysis observed in cancer and stressed myocardium; produces metabolites that influence epigenetic modifiers. |

(e.g., non-coding RNAs) is included. Although transgenerational studies of non-Mendelian traits (e.g., cardiometabolic disease) support the existence of Neo-Lamarckian inheritance patterns via epigenetic transference (Loison, 2021), this review does not delve into heredity and focuses instead on the molecular mechanisms of adult-onset disease conferred via epigenetic machinery. Specifically, we highlight the functional relevance of chromatin-modifying enzymes (i.e., DNA- and histone-modifying enzymes) that are known to influence either accessibility or activity states of genes (Bird, 2007).

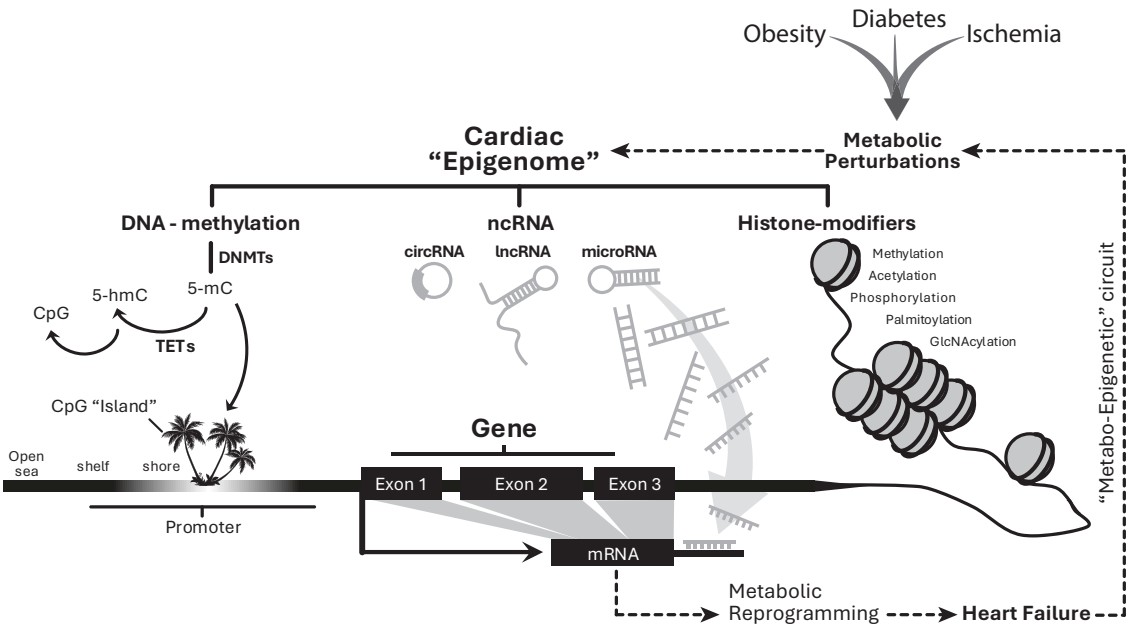

**Figure 1. Epigenetic mechanisms as regulatory features of metabolic gene programs in heart failure.**

DNA methylation can be covalently added to cytosine moieties via DNA methyltransferases (DNMTs) and removed via ten-eleven translocases (TETs), with regions of methylation dynamics centered around CpG regions based on CpG content. Non-coding RNAs (ncRNA) includes long non-coding and microRNAs and classically affects mRNA stability and/or degradation post-transcriptionally. Lastly, histone-modifying enzymes covalently add a variety of functional groups to modulate transcription factor affinity and/or gene accessibility by influencing chromatin folding and macrostructure.

Emerging evidence points to the functional relevance of chromatin-modifying enzymes as both direct and indirect regulators of cardiac metabolism. Conversely, the failing heart undergoes significant epigenetic and metabolic changes including intermediary metabolites even before the onset of cardiac structural or functional impairment (Razeghi et al, 2001; Kundu et al, 2015), illustrating a conceptual framework whereby environmental perturbations leverage existing 'metabo-epigenetic circuits' to dysregulate cardiac function (Fig. 2).

In this review, we summarize established and emerging concepts pertaining to the role of cardiac metabolism and epigenetics in the pathogenesis of cardiomyopathy. Specifically, we highlight evidence that 'metabo-epigenetic circuits' integrate metabolic stressors into the transcriptional signals that influence cardiac function and how chromatin modifying enzymes regulate metabolic programs. Missing nodes emerge from within these signaling networks, the discovery of which would manifest therapeutic opportunities to rewrite the metabolic and epigenetic origins of heart disease.

## A synopsis of cardiac metabolic plasticity

Several transitions occur throughout cardiac development and into adulthood that influence metabolic signatures. The healthy adult heart exhibits a clear preference for fatty acid oxidation (FAO), yet it is not dependent on FAO to sustain baseline cardiac function. Limiting fatty acid uptake via cardiac-specific knockout of CD36 (fatty acid translocase) does not cause cardiac dysfunction (Nagendran et al, 2013). Similarly, mice lacking cardiac peroxisome

proliferator-activated receptor alpha (PPARα$^{-/-}$) exhibit normal structural and hemodynamic properties in the short-term; however, these mice eventually develop cardiac structural abnormalities reminiscent of human dilated cardiomyopathy when subjected to chronic PPARα genetic disruption in aged (9-month-old) mice (Guellich et al, 2007). It has become apparent in these studies, therefore, that the consumption of alternative fuels accommodates reduced fatty acid metabolism, balancing the heart's cumulative metabolic demands (Kassiotis et al, 2008). Consequently, fatty acid metabolism is not required to maintain baseline cardiac function owing to the metabolic reserve that is afforded by the heart's alternate fuel sources.

In contrast to the adult heart, the fetal heart exhibits a distinct metabolic profile and circulatory physiology. Beginning around 8 weeks of gestation, fetal circulation is maintained by parallel cardiac outputs from both ventricles through in utero shunts that bypass the pulmonary circuit (Doubilet and Benson, 1995). Owing to the low-resistance of placental circulation, fetal mean arterial pressures are significantly lower than in adults, typically ranging from 25–45 mmHg (Struijk et al, 2008), with ejection fractions between 60 and 90% (Simioni et al, 2011). Owing to low oxygen tension—attributable to the high oxygen affinity of fetal hemoglobin (Allen et al, 1953)—the fetal heart thrives on abundant glucose and lactate (Fisher, 1984; Lopaschuk et al, 1992). This preference is supported by elevated expression of key glycolytic enzymes and transporters (GLUT1, HK1, PFK1, LDHA) (Razeghi et al, 2001) and reduced expression of fatty acid oxidation genes and regulators such as CPT-1b, MCD, PGC-1α, and PPARα (Lopaschuk et al, 1992).

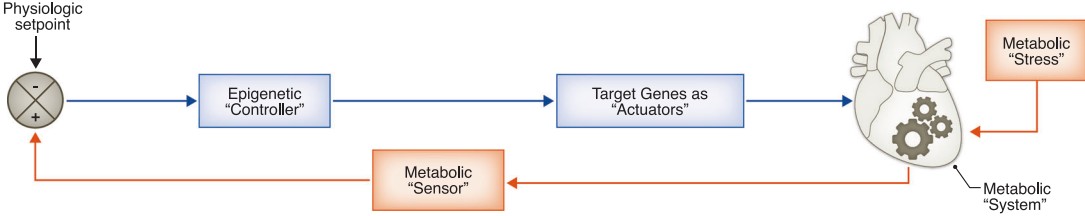

**Figure 2. A conceptual framework of "metabo-epigenetic" signaling in the heart.**

Metabolic perturbations act as "sensors" that generate molecular signals influencing myocardial gene expression ("actuators") through epigenetic regulatory mechanisms ("controllers"), forming a feedback loop analogous to a proportional–integral–derivative (PID) control system.

## Metabolic perturbations in dilated cardiomyopathy

When it fails, the heart initially compensates by reactivating a metabolic program characteristic of fetal hearts (Taegtmeyer et al, 2010; Komuro and Yazaki, 1993), increasing glycolysis and ketone body utilization (Oka et al, 2015; Doenst et al, 2013; Aubert et al, 2016). This process of metabolic substrate switching has been recognized for nearly half a century, wherein the failing adult heart increases glucose utilization for energetic supply while simultaneously lowering fatty acid oxidation (Bishop and Altschuld, 1970). The metabolic derangements of heart failure are now known to precede both the morphological changes and clinical manifestations of heart failure. Non-invasive metabolic assessments of patients with hypertension have since uncovered that, even before the onset of ventricular dysfunction, glucose utilization is augmented relative to non-hypertensive patients (Hamirani et al, 2016). Metabolomics of circulating metabolites confirmed that consumption of ketones and lactate increases in heart failure (Murashige et al, 2020). For this reason, the traditional concept that the failing heart lacks fuel for energy production (Neubauer, 2007) has lost favor, shifting instead towards an understanding that metabolic substrates employed for energy production act as regulatory intermediates to modulate cardiac performance.

## Cardiac metabolic perturbations of cardiometabolic disease

The prevalence of obesity, diabetes and heart failure has exceeded epidemic proportions across Western society (Braunwald, 1997; Mant et al, 2009; Roger, 2013), accompanied by rising rates of end-organ complications such as cardiomyopathy. While diabetes often coexists with traditional risk factors like coronary artery disease and arterial hypertension, it independently confers a fourfold increased risk of heart failure (Mahmood and Wang, 2013). Diabetic patients are at a greater risk of hospitalization and cardiovascular-related mortality (Dei Cas et al, 2015; Sarma et al, 2013), and show a diminished response to guideline-directed therapies for heart failure (HF) with reduced ejection fraction (HFrEF) (Dei Cas et al, 2015). Heart failure attributable to metabolic syndrome—often termed cardiometabolic heart failure with preserved ejection fraction (HFpEF)—typically involves features of both systolic and diastolic dysfunction (Ritchie and Abel, 2020; Zarich and Nesto, 1989). Notably, up to half of asymptomatic individuals with diabetes exhibit subclinical signs of

myocardial dysfunction (Marwick et al, 2018; Boyer et al, 2004). Tighter glycemic control does not improve outcomes in HFpEF, and intensive glucose lowering may actually worsen cardiovascular outcomes (Lejeune et al, 2021; Shi et al, 2024).

Distinct from dilated cardiomyopathies, HFpEF exhibits unique metabolic and functional derangements (Maack et al, 2018). While failing hearts often increase glucose and ketone metabolism (Oka et al, 2015; Doenst et al, 2013; Aubert et al, 2016), diabetic hearts rely heavily on fatty acid oxidation, driven by insulin resistance and impaired glucose uptake (Shao and Tian, 2015; Rodrigues et al, 1995; Pepin et al, 2025). This observation has motivated functional studies seeking to understand the consequences of the diabetic heart's persistent reliance on fatty acids. In this effort, Umbarawan et al demonstrated that impaired fatty acid uptake via FABP4/5 deficiency worsens cardiac dysfunction in diabetic mice (Umbarawan et al, 2020), while increased lipolysis through *Pnpla2* (ATGL) overexpression can be protective (Kienesberger et al, 2012, 2013). However, excessive lipid accumulation disrupts homeostatic signaling events in subcellular compartments of the mitochondrion, cytosol, and nucleus (Haemmerle et al, 2006, 2011; Wende and Abel, 2010).

Efforts to restore glucose utilization have thus far yielded mixed results. While cardiomyocyte-specific *Glut4* expression failed to improve heart function in diabetic mice (Wende et al, 2020), overexpression of *Glut1* reduced fatty acid oxidation but impaired contractile reserve (Yan et al, 2009). Additionally, Brahma et al showed that diabetic hyperglycemia suppresses ketone metabolism by inhibiting proteins 3-hydroxybutyrate dehydrogenase 1 (BDH1) and 3-oxoacid CoA-transferase 1 (OXCT1) via O-GlcNAcylation, revealing complex substrate competition in the diabetic heart (Brahma et al, 2020).

## Rare human genetic cardiomyopathies highlight metabolic origins of heart failure

Fine mapping and mechanistic validation of rare human genetic variants together reinforce a metabolic axis of dilated cardiomyopathy. Genetic contributions are estimated to account for roughly 70% of the lifetime risk for heart disease, even after adjustment for modifiable lifestyle factors and comorbid illnesses (Lee et al, 2006). Although most of these common variants affect lipid metabolism, other monogenic cardiomyopathies occur solely via loss-of-function mutations within metabolic genes (Table 1). In the following section, we summarize the mechanistic insights

**Table 1. Monogenic etiologies of metabolic cardiomyopathy.**

| Gene | Molecular function | Cardiac consequence | Clinical syndrome | Reference |
|---|---|---|---|---|
| MT-TL1 | mt-tRNA defects → reduced OXPHOS expression | HCM, DCM | MELAS | (Hsu et al, 2016; Burattini et al, 2024) |
| MT-TK | Mitochondrial tRNA-Lys (mt-protein synthesis) | HCM, DCM | MERRF | (Catteruccia et al, 2015) |
| POLG1 | Mitochondrial DNA polymerase (mtDNA replication/repair) | ACM, DCM | Mitochondrial DNA depletion syndrome (Alper's disease) | (Spracklen et al, 2021) |
| ACAD9 | Complex I assembly factor and acyl-CoA dehydrogenase (fatty acid β-oxidation) | DCM | ACAD9 deficiency | (Dewulf et al, 2016) |
| ACADVL | Very long-chain fatty acid β-oxidation | HCM, Lipid accumulation | VLCAD deficiency | (Parini et al, 1998) |
| AARS2 | Mitochondrial alanyl-tRNA synthetase (mt-protein synthesis) | HCM | infantile mitochondrial cardiomyopathy | (Zhang et al, 2024) |
| COX15 | Heme A biosynthesis (cytochrome c oxidase assembly, complex IV) | Early-onset fatal HCM | Leigh syndrome | (Antonicka et al, 2002) |
| CPT2 | Carnitine palmitoyltransferase II (fatty acid transport) | HCM, Lipid accumulation | CPT2 deficiency | (Pereyra et al, 2017) |
| ELAC2 | Mitochondrial RNase Z (tRNA 3' end processing, mtRNA maturation) | Severe infantile-onset HCM | Combined oxidative phosphorylation deficiency 17 | (Saoura et al, 2019) |
| ETFA/B/DH | Electron transfer flavoprotein (fatty acid/amino acid oxidation) | HCM, Lipid Accumulation | Multiple acyl-CoA dehydrogenase deficiency (MADD) | (Singla et al, 2007) |
| FKRP | Ribitol-5-phosphate transferase (glycosylation of α-dystroglycan) | DCM | FKRP-CDG (dystroglycanopathy) | (Libell et al, 2020) |
| HADHA | Long-chain 3-hydroxyacyl-CoA dehydrogenase (fatty acid β-oxidation) | HCM, DCM | LCHAD/Trifunctional protein deficiency | (Gaston et al, 2023) |
| GLA | α-galactosidase A (lysosomal glycosphingolipid degradation) | HCM | Fabry disease | (Weissman et al, 2024) |
| LAMP2 | Lysosomal-associated membrane protein 2 (autophagy) | HCM | Danon disease | (Nishino et al, 2000) |
| GAA | Acid α-glucosidase (glycogen hydrolysis) | HCM | Pompe disease (glycogenosis type II) | (Taverna et al, 2020) |
| PRKAG2 | AMP-activated protein kinase γ2 (energy sensing) | HCM | PRKAG2 syndrome | (Ahamed et al, 2020) |
| AGK | Acylglycerol kinase (phospholipid metabolism) | HCM | Sengers syndrome | (Das et al, 2019) |
| DPM3 | Dolichol-phosphate mannosyltransferase (N-glycosylation) | DCM | Congenital disorder of glycosylation (MDDGC15) | (Svahn et al, 2019) |
| PNPLA2 | Hydrolyzes cytoplasmic triacylglycerol (TAG) in lipid droplets (lipolysis) | DCM, HCM | Neutral lipid storage disease with myopathy (NLSDM) | (Wang et al, 2024) |
| ABHD5 | Serine protease (links lipid and glucose metabolism via HDAC4-NT production) | DCM | Chanarin-Dorfman Syndrome (CDS) | (Mangukiya et al, 2023) |

Variants within metabolic genes known to cause cardiopathy are listed, along with the molecular function and clinical syndrome.
ACM arrhythmogenic cardiomyopathy; HCM hypertrophic cardiomyopathy; DCM dilated cardiomyopathy; MELAS mitochondrial encephalomyopathy, lactic acidosis, and stroke-like episodes; MERRF myoclonus epilepsy with ragged-red fibers.

incidentally gained from familial monogenic dilated cardiomyopathies harboring metabolic intermediate gene mutations.

Pathogenic variants associated with *PNPLA2* (well known as ATGL), the enzyme that catalyzes lipolysis to hydrolyze TAGs to DAGs to liberate intracellular lipid droplets (Smirnova et al, 2006), produce a severe cardiac phenotype leading to heart failure, termed neutral lipid storage disease with cardiomyopathy (NLSD-CM) (Fischer et al, 2007; Schweiger et al, 2009; Pennisi et al, 2017). The clinical manifestations of NLSD-CM depend on the type of mutation and on residual enzymatic activity (Schweiger et al, 2009), but it has also been theorized to occur from alterations in chromatin accessibility (Pennisi et al, 2017). The clinical phenotype of human NLSD-CM resembles that of *Pnpla2*-deficient mice, wherein cardiac lipid droplet accumulation and reduced fatty acid oxidation (FAO) accompany progressive cardiac dysfunction and eventually decompensated heart failure (Haemmerle et al, 2006, 2011). In mice, ventricular systolic function can be corrected by both overexpressing *Pnpla2* and treatment with PPARα agonist (Haemmerle et al, 2011).

As a coactivator of PNPLA2, ABHD5 (well known as CGI-58) has also been shown to produce similar phenotypic manifestations when genetically disrupted in mice, although far fewer clinical cases of genetic *ABHD5* mutations have been found (Pennisi et al, 2017). Constitutive and cardiomyocyte-specific knockout of *Abhd5* in mice produces a severe cardiomyopathy (Jebessa et al, 2019). The latter study revealed that—other than FAO and lipotoxicity—a tandem loss of HDAC4-NT (an N-terminal proteolytic fragment of histone deacetylase 4, which is produced by a yet undiscovered proteolytic activity of Abhd5) accounts for the functional phenotype in mice. Mechanistically, the loss of HDAC4-NT activates a distinct transcriptional program via transcription factor Myocyte Enhancer Factor 2 (MEF2) that activates nuclear receptor subfamily 4 group A member (NR4A1), a key enzyme that activates the hexosamine biosynthetic pathway (HBP) and subsequent protein O-GlcNAcylation of calcium handling proteins, in particular a specific splicing variant of stromal interaction molecule 1 (STIM1-L) (Lehmann et al, 2018). Thus, the discovery that HDAC4-NT, which itself is sufficient to rescue cardiac dysfunction by cardiac gene therapy in *Abhd5*-deficient mice despite persisting lipid droplet accumulation, challenges the concept of lipotoxicity and directs the attention to pathological accumulation of intermediates and byproducts of glucose metabolism as drivers of transcriptionally-driven cardiac dysfunction. HDAC4-NT critically regulates the production of these glycolytic byproducts. This finding necessitates further studies that weigh the relative contributions of lipotoxicity versus glucotoxicity in both NLSD-CM and other, more common cardiometabolic etiologies of heart failure.

## Metabo-epigenetic sensors

Metabolic intermediates serve not only as fuels but also as signaling molecules that influence chromatin-modifying enzymatic activity. In 1924, Otto Warburg first reported that cancer cells prefer anaerobic glycolytic metabolism, or "fermentation," even when exposed to oxygen-rich environments, a phenomenon that later became coined as the Warburg Effect (Warburg, 1956). Many of the metabolic alterations that are characteristic of malignancies, such as changes in TCA cycle flux and substrate preference, mirror those found in the failing heart (Taegtmeyer et al, 2017). In this manner,

an array of metabolic intermediates and byproducts collectively defined as "oncometabolites" (Yang et al, 2013) have also been studied as signaling molecules that promote heart failure pathogenesis.

In non-cardiac tissue, TCA cycle intermediates including succinate and fumarate have been shown to influence chromatin-modifying enzymes. Mitochondrial accumulation of both succinate and fumarate via loss-of-function mutations of succinate dehydrogenase (SDH) or fumarate hydratase (FH) blocks the activity of ten-eleven translocase (TET) dioxygenases to inhibit nuclear DNA demethylation and JmjC domain-containing histone demethylases (KDM), thus de-repressing tumor suppressor and metabolic genes (Xiao et al, 2012). Similarly, gain-of-function mutations in tumor-derived isoforms of isocitrate dehydrogenase (IDH1 and IDH2) drive the accumulation of D-2-hydroxyglutarate, a competitive TET and KDM inhibitor, and blocks both histone demethylation and 5-hydroxymethylcytosine oxidation (Waitkus et al, 2018; Xu et al, 2011; Chowdhury et al, 2011).

In the heart, expression of mutant IDH2 in mice leads to D-2-hydroxyglutarate accumulation, impaired oxidative capacity, and cardiomyopathy (Karlstaedt et al, 2016; Akbay et al, 2014). These metabolic changes are accompanied by alterations in histone acetylation and methylation, supporting a causal role for metabolic-epigenetic coupling in cardiac dysfunction (Taegtmeyer et al, 2017). Thus, although not fully characterized in the heart, these findings support the framework that metabolic intermediates reprogram gene expression by altering modifiers of histone and DNA methylation.

Beyond the TCA cycle, the products of glucose metabolism also confer epigenetic control of gene expression via protein O-GlcNAcylation (see also above in the section "Rare human genetic cardiomyopathies highlight metabolic origins of heart failure"). O-GlcNAcylation is known to affect tumor behavior by enhancing HDAC1 activity (Zhu et al, 2012) and modifying signaling pathways that drive malignancy (Chu et al, 2020; Wu et al, 2020). In the heart, elevated O-GlcNAcylation has been implicated in diabetic cardiomyopathy and arrhythmogenesis (Jin et al, 2020), though its effects appear context-dependent (Lunde et al, 2012; Zhu et al, 2019). A recent study demonstrated that mitochondrial dislocation of hexokinase 1 (HK1) in endothelial cells promotes its interaction with O-linked N-acetylglucosamine transferase (OGT), increasing protein O-GlcNAcylation and driving HFpEF pathogenesis (Tatekoshi et al, 2025). Deletion of the mitochondrial binding domain in HK1 led to spontaneous HFpEF with impaired angiogenesis, while reversal of O-GlcNAcylation restored vascular and diastolic function.

Taken together, these observations underscore O-GlcNAcylation as a metabolically sensitive epigenetic moiety. The modification of histone deacetylases by O-GlcNAc, particularly protein O-GlcNAcylation at serine 642 of HDAC4 (see more details below in the section "HDACs"), is unmasked to protect against maladaptive signaling through CAMKII in diabetic hearts (Kronlage et al, 2019). Given the association of protein O-GlcNAcylation with both adaptive and maladaptive cardiac remodeling (Dubois-Deruy et al, 2015; Lunde et al, 2012; Muthusamy et al, 2014), additional mechanistic studies are necessary to dissect the site-specific and context-dependent effects of this modification to determine its role in heart failure pathogenesis.

**Table 2. Summary table of known metabo-epigenetic circuits.**

| Stress | Sensor | Controller | Gene program | System variable | Species | Reference(s) |
|---|---|---|---|---|---|---|
| Glucose handling | | | | | | |
| Exercise | ↑ PKA / ↑ ABHD5 | ↑ HDAC4-NT | ↓ Nr4a1 | ↓ HBP, ↑ Contractility | Mouse | (Lehmann et al, 2018) |
| Pressure-Overload | ↑ CAMK2D | ↓ HDAC4-NT | ↑ Nr4a1 → ↑ Gfpt2 | ↑ HBP → ↑ STIM1L O-GlcNAc ↓ Contractility | Mouse | (Lehmann et al, 2018) |
| Pressure-Overload | ↓ ABHD5 | ↓ HDAC4-NT | ↑ Nr4a1 → ↑ Gfpt2 ↑ Pdk4 → ↑ PDH-phos. | ↓ Glycolysis, ↑ HBP (O-GlcNAc) ↓ Contractility | Mouse | (Jebessa et al, 2019) |
| Diabetes (STZ, Db/Db) | ↑ HBP (O-glcNAc) | ↑ HDAC4-NT | ↓ Nr4a1 | ↓ HBP, ↑ Contractility | Mouse | (Kronlage et al, 2019) |
| – | – | ↓ DNMT3A | ↓ MYH7/MYH6 ↑ PPARγ | ↓ Glycolysis, Lipid accumulation | In Vitro (iPSCs) | (Madsen et al, 2020, 2021) |
| Diabetes (STZ) | N/A | N/A | "stable" | N/A | Mouse | (Lother et al, 2020) |
| ICM | – | ↑ miRNA-320 | ↓ Pfk, ↓ Hsp20 | ↑ Glycolysis ↓ Infarct Size | Human & Mouse | (Ren et al, 2009) |
| ICM | – | ↑ EZH2 | ↑ Pfkfb3, ↓ Klf15, ↓ Acadvl/m/sb ↓ Idh3a, ↓ Sdhb, ↓ Cox5a/b | ↑ Glycolysis ↓ FAO | In vitro & Human | (Pepin et al, 2019b) |
| Oxidative phosphorylation | | | | | | |
| Pressure-Overload | IDH1/2 → D-2-hydroxy-glutarate | Histone 3 KDMs/HATs | – | ↓ OX-PHOS | Mouse | (Karlstaedt et al, 2016) |
| DCM | – | ↑DNMT3A ↓GADD45B ↑HDAC4/7 | ↓NRF1 target genes, ↓Acsl5, ↓Acsl1, ↓Hadha, ↓Ndufa5, ↑Pfkfb3, ↑Eno2 | ↑ Glycolysis ↓ FAO | In vitro (H9c2) | (Pepin et al, 2019a) |
| Diabetes (db/db) | – | miRNA-320 | ↑Ago2 → ↑Cd36 | ↑ Fatty Acid Uptake | Mouse | (Li et al, 2019) |
| Circuits controlling different aspects of cardiac functions | | | | | | |
| Dietary Cholesterol | – | miRNA-25 | ↑ Nox4 | ↑ Diastolic Dysfunction | Mouse | (Varga et al, 2013) |
| DCM | – | – | ↑ AMOTL2 ↑ ARHGAP2 ↓ PECAM1 | ↑ Angiogenesis | Human | (Movassagh et al, 2010) |
| – | – | Sirt6+/- | FoxO1 → ↑ Pdk4 ↓ Pdh | ↓ Oxygen Consumption | Mouse | (Khan et al, 2018) |
| – | – | Hdac3-\ | ↑ FA Uptake, ↑ FAO, ↑ OX-PHOS, ↑ Ppara | Cardiac Hypertrophy, Lipid Accum. | Mouse | (Montgomery et al, 2008) |

# Known metabo-epigenetic circuits

The interface between metabolism and epigenetic regulation is increasingly recognized as a contributor to heart failure pathogenesis. Below, we summarize the current evidence supporting chromatin modifiers (specifically histone acetylation and DNA methylation) as regulatory links between nutrient sensing and transcriptional control of metabolic intermediate genes. These metabo-epigenetic circuits offer insight into how metabolic states are encoded at the chromatin level, though many remain incomplete, lacking defined sensors, regulators, or target genes (Table 2), and unmask new and promising therapeutic targets.

## HDACs

Histone deacetylases (HDACs) are key regulators of cardiac metabolism that link gene accessibility to metabolic gene expression to control myocardial energy homeostasis. Traditionally recognized for their roles in chromatin compaction and transcriptional repression, HDACs are now appreciated for their dynamic influence on myocardial metabolic flexibility, substrate preferences, and heart failure pathogenesis.

Acetyl-CoA, generated from glucose, glutamine or fatty acids via ATP-citrate lyase (ACL), from either pyruvate by pyruvate dehydrogenase (PDH) or acetate via acetyl-CoA synthetase (ACSS), provides the substrate for histone acetylation and thus provides an important link between energy metabolism and chromatin regulation (Wellen and Thompson, 2012; Rathmell and Newgard, 2009). In a non-cardiac (HCT116) colorectal carcinoma cell line, high-glucose media increases histone acetylation, whereas reduction in acetyl-CoA synthesis by blocking ACLY results in rapid histone deacetylation (Wellen et al, 2009). Although supplementation with fatty acids complements the cells' bioenergetic needs during glucose deprivation, no measurable effect is observed on histone acetylation (Wellen et al, 2009). Similarly, the acetylation of several metabolic intermediate proteins (e.g., GLUT4, HK2, PFK1 and IDHA) in

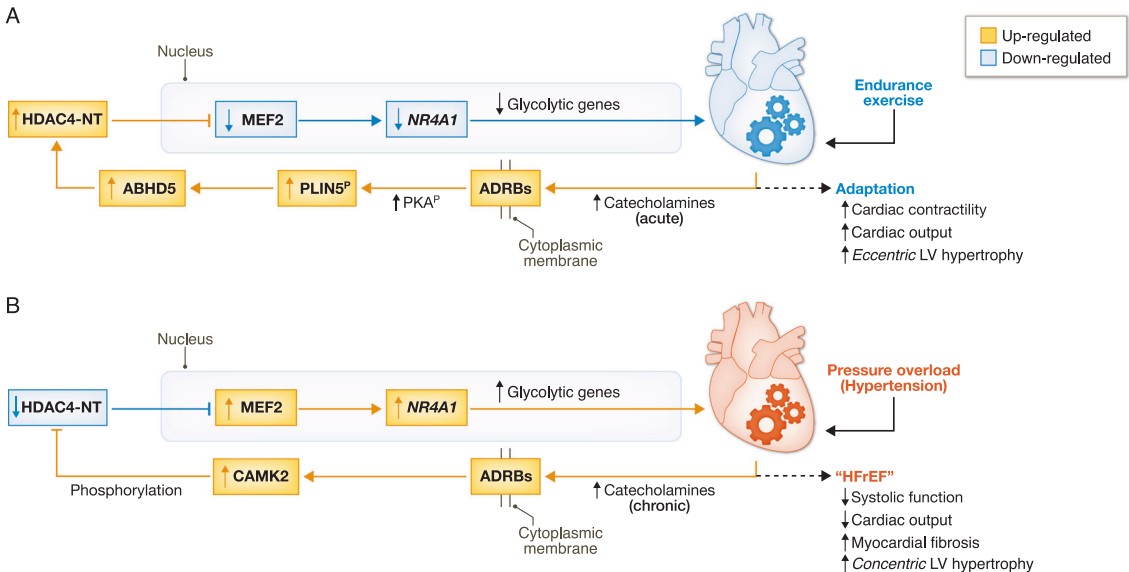

**Figure 3. Metabo-epigenetic circuits involving HDAC4.**

(A) Physiologic response to intermittent catecholamine exposure (e.g., endurance exercise), promoting adrenergic signaling-mediated enhancement of nuclear HDAC4-NT suppression of MEF2, facilitating an adaptive metabolic response. (B) chronic catecholamine signaling (e.g., pressure overload, hypertension) drives pathological metabolic remodeling via CaMK2-mediated suppression of HDAC4 translocation causing de-repression of glycolytic genes.

differentiated adipocytes is regulated in an ACL-dependent manner in accordance with glucose availability (Wellen et al, 2009).

Although global HDAC inhibition (panHDACi) mitigates the pathological features of heart failure in mice (Travers et al, 2021), class I and class II HDACs seem to control this regulatory network through distinct—yet overlapping—mechanisms, modulating transcription factor activity, metabolic enzyme expression, and post-translational modifications in a context-dependent manner. The following section explores the roles of individual HDAC isoforms as metabo-epigenetic "controllers" in the heart, showcasing their diverse contributions to cardiac function and metabolic remodeling in heart failure.

### Class I

Among class I HDACs, HDAC3 has been identified to maintain cardiac energy metabolism (Montgomery et al, 2008). *Hdac3*-deficiency in mice leads to massive cardiac hypertrophy along with upregulation of genes associated with fatty acid uptake, fatty acid oxidation, and oxidative phosphorylation, resulting in fatty acid-induced myocardial lipid accumulation and elevated triglyceride levels (Paluvai et al, 2023). As a potential underlying mechanism, excessive activity of the nuclear receptor Peroxisome proliferator-activated receptor alpha (PPARα) is seen, placing HDAC3 within this metabo-epigenetic circuit. Because this phenotype differs from other class I HDAC knockout phenotypes, a unique role for HDAC3 in the maintenance of cardiac function and regulation of myocardial energy metabolism was assumed until only recently. However, given the recently identified phenotypic overlap of murine *Hdac4* and *Hdac3*-deficiency in the heart, it will be interesting to investigate the functional relationship of HDAC3 and HDAC4 to regulate cardiac metabolism. This is of particular interest because it is known for more than two decades that these

HDACs physically interact via the transcriptional corepressor N-CoR/SMRT (Fischle et al, 2002), and thus may jointly regulate cardiac metabolism with HDAC4 serving as the signal-responsive partner.

### Class IIa and class IIb

Class II HDACs consist of class IIa (HDAC4, HDAC5, HDAC7 and HDAC9) and class IIb HDACs (HDAC6 and HDAC10). Among these, it has been long recognized that class IIa HDAC enzymes participate in cardiac metabolic homeostasis (Zhang et al, 2002; Backs et al, 2006; Chang et al, 2004; Hsu et al, 2020; Schuster et al, 2015). These studies have shown that, in response to pressure overload, class IIa HDACs act as signal-responsive regulators of cardiac growth via non-enzymatic mechanisms resulting in MEF2 inhibition (McKinsey et al, 2000; Zhang et al, 2002; Backs et al, 2006; Chang et al, 2004). MEF2 tightly controls the expression of NR4A1 (Youn et al, 1999), which has been shown to both regulate the expression of several metabolic enzymes (Pearen and Muscat, 2010) and specifically increase the activity of the HBP (Lehmann et al, 2018).

Our prior work has highlighted a decisive role of HDAC4 via *Nr4a1* regulation as part of a transcriptional program that leads to impaired cardiac calcium homeostasis leading to systolic dysfunction (Lehmann et al, 2018) (Fig. 3). The HDAC4-MEF2-NR4A1 pathway is controlled by the proteolytic activity of the lipid droplet-associated protein ABHD5. Specifically, ABHD5 is downregulated in response to transaortic constriction (TAC) along with the product of its proteolytic activity: HDAC4-NT, leading to MEF2-dependent stimulation of inefficient cardiac glycolysis with accumulation of UDP-GlcNAc and subsequent protein O-GlcNAcylation and impaired cardiac performance (Jebessa et al, 2019).

Among the Class IIb family of HDAC enzymes, HDAC6 has attracted the most attention as a pharmaceutical candidate since its selective inhibition (e.g., via TYA-018/TN-301) lowers diastolic dysfunction in a dual high-fat diet and transaortic constriction model of HFpEF (Ranjbarvaziri et al, 2024). HDAC6 inhibition is also associated with enhanced post-translational acetylation of microtubule filaments and the sarcomeric constituent titin, which is thought to stabilize cytoskeletal integrity, reduce inflammatory signaling, and improve mitochondrial function (Lin et al, 2022). These data are puzzling, however, since global *Hdac6*-deficient (*Hdac6*-KO) mice develop worsening diastolic dysfunction (Lin et al, 2022), suggesting that baseline HDAC6 activity may be essential to maintain normal cardiac compliance. Future studies must define the dose-dependent cellular and molecular mechanisms responsible to resolve this apparent contradiction. Conditional *Hdac6* knockout studies may also be useful to identify the responsible cell type using in vivo models, and domain-specific perturbations are needed to differentiate between enzymatic and non-enzymatic functions of HDAC6. Moreover, as tubulin deacetylation has been suggested as the target of HDAC6, functional studies should gain a better understanding of HDAC6 in HFpEF regarding how tubulin deacetylation mitigates diastolic dysfunction in HFpEF. Thus, while HDAC6 inhibition shows therapeutic potential, defining its potential role in HFpEF pathogenesis warrants further study of the cell-type specific, dose-dependent, and context specific regulatory balance of epigenetic regulation and metabolic signaling.

### Class III (sirtuins)

Sirtuins (SIRTs) comprise a family of seven NAD$^+$-dependent deacetylases, among which several are heralded as epigenetic controllers of cardiac metabolic substrate sensing and gene regulation. Specifically, SIRT1, SIRT3, and SIRT6 play distinct yet interconnected roles in modulating myocardial substrate preferences, orchestrating a balance between glucose, fatty acid, and ketone metabolism.

SIRT1 is mostly found in the nucleus and regulates cardiac energy metabolism through dynamic interactions with—and deacetylation of—the transcription factors PPARα, Peroxisome Proliferator-Activated Receptor Gamma Coactivator 1-alpha (PGC-1α), and AMP-activated protein kinase (AMPK) to indirectly regulate mitochondrial biogenesis, FAO, and glucose utilization (Zhou et al, 2024). In diabetic cardiomyopathy (clinically termed cardiometabolic HFpEF), recombinant SIRT1 supplementation improves myocardial function by suppressing PPARγ signaling, reducing lipid accumulation, and downregulating genes involved in lipid trafficking and inflammation (Majeed et al, 2021). Its activation enhances oxidative metabolism (Rodgers et al, 2005), whereas reduction of SIRT1 promotes the development of dilated cardiomyopathy (Gorski et al, 2019; Prola et al, 2017).

Unlike other SIRT members, SIRT3 localizes to the mitochondrial matrix and functions in situ as a mitochondrial stress sensor (Chen et al, 2021). It post-translationally deacetylates—and thus activates—enzymes essential for FAO, ketogenesis, and oxidative phosphorylation to promote efficient ATP production and redox homeostasis (Chen et al, 2015). Conversely, *Sirt3* deficiency accelerates lipotoxicity, oxidative stress, and energetic failure. In diabetic and aging hearts, the downregulation of *Sirt3* is associated with impaired metabolic flexibility and heart failure progression; its

restoration normalizes FAO and ketone metabolism to mitigate cardiac hypertrophy and fibrosis (Chen et al, 2021; Palomer et al, 2020). Similarly, nutritive supplementation with the NAD+ precursor nicotinamide riboside may mitigate diastolic dysfunction and cardiomyocyte senescence in HFpEF (Tong et al, 2021). However, recent data have challenged the central role of SIRT3 in conferring a cardioprotective role of NAD+, since nutritive restoration of NAD+ also appears to be effective in *Sirt3*-knockout mice. Specifically, by comparing the relative effects of supplementation with nicotinamide riboside in mice subjected to pressure-overload induced cardiomyopathy, Walker et al found that there were similar benefits on NAD(H) redox-sensitive short-chain dehydrogenase/reductase function and restoring mitochondrial oxidative metabolism in both *Sirt3*-deficient and wild-type mice (Walker et al, 2023). Additional studies are thus needed to clearly define the SIRT3-dependent and independent mechanisms of the NAD-associated therapeutic benefits in HFpEF.

SIRT6 acts within the nucleus to regulate expression of genes encoding lipid handling and inflammation, particularly in the diabetic heart. SIRT6 suppresses transcription of lipid transporters such as CD36 and FABP4 by occupying the DNA-binding domain of PPARγ and preventing its binding to its target promoters (Khan et al, 2021), thereby reducing fatty acid uptake and preventing lipotoxic remodeling (Wu et al, 2022). Pharmacologic activation (e.g., MDL-800) has shown promise in preclinical models of HFpEF by restoring lipid homeostasis and reducing myocardial inflammation (Wu et al, 2022).

Together, these studies support the theory that sirtuins serve as key regulators within metabo-epigenetic circuits to link nutrient availability, chromatin remodeling, and cardiac metabolic gene programming in both adaptive and pathological contexts.

## DNA methylation modifiers

Our group and others have shown that metabolic substrate switching in human HFrEF is encoded by alterations in DNA methylation. Genome-wide methylation profiling of failing human cardiac biopsies uncovered hypermethylation of oxidative metabolic gene promoters that is coupled to hypomethylation of glycolytic gene promoters, collectively recapitulating a fetal-like metabolic return of glycolytic energy metabolic preference (Pepin et al, 2019a). DNA methyltransferase 3A (DNMT3A) in human induced pluripotent stem cell-derived cardiomyocytes (iPSC-CMs) is essential for maintaining metabolic homeostasis in human cardiomyocytes (Madsen et al, 2020), while genetic overexpression of *DNMT3A* in rat cardiomyoblasts suppresses expression of gene intermediates including *ACSL1* and *HADHA* (Pepin et al, 2019a). Conversely, *DNMT3A* knockout in human engineered heart tissue causes myocardial lipid accumulation via PPARγ activation and impaired glucose metabolism due to destabilization of HIF-1α. Together, these studies highlight a causal link of de novo DNA methyltransferase activity in coordinating both structural and metabolic integrity of differentiated cardiomyocytes (Madsen et al, 2021).

The oxidative reaction of actively removing DNA methyl groups is catalyzed by ten-eleven translocation (TET) enzymes, which have similarly been linked to myocardial metabolic plasticity. TET deficiency disrupts cardiac development by inducing hyperactivation of WNT signaling and suppressing NKX2-5, a transcription

factor that is essential for cardiomyocyte lineage commitment (Lan et al, 2021). Clinically, DNA methylation changes correlate with suppressed mitochondrial gene expression and enhanced glycolysis in HF patients (Pepin et al, 2019a). Mouse models of *Tet2*-deficient clonal hematopoiesis exhibit accelerated HF progression due to IL-1β-mediated inflammation (Sano et al, 2018), suggesting that hematopoietic cell-based epigenetic changes indirectly alter myocardial metabolic homeostasis. Together, these studies define DNA methylation as a metabo-epigenetic circuit, providing preliminary insights into how metabolic stress, developmental programs, and inflammation are encoded by methylome-based metabo-epigenetic regulatory gene programs.

Regarding the impact of the diabetic milieu on cardiac DNA methylation, a recent study has suggested that, although cardiac gene expression is altered in streptozotocin-induced diabetic mice, cardiac DNA methylation appears "stable," or unaffected (Lother et al, 2020). This finding falls contrary to candidate gene studies that highlight differential DNA methylation as a measurable stress signal in diabetes (Liu et al, 2020; Chen et al, 2019). Ultimately, sufficiently powered studies are required to understand whether differential DNA methylation occurs due to the direct metabolic stress of diabetes in the heart.

# Myocardial pharmaco-epigenetic basis of metabolic therapies

Owing to the central importance of metabolic derangements in the development of heart disease, it is sensible to theorize that targeting these metabolic disruptions would remedy the hemodynamic and cardiac functional consequences that culminate into clinical heart failure (Koutroumpakis et al, 2020). Although precision-based therapies have not yet yielded a pharmacologic treatment designed to alter cardiac metabolism, several existing cardioprotective agents influence cardiac metabolism. Additionally, the effectiveness of any targeted epigenetic therapy is likely affected by guideline-directed medical therapies (GDMT) for heart failure, which themselves influence metabolic gene accessibility. For this reason, below we highlight the current understanding regarding the metabolic and epigenetic effects of existing HF therapies. It is worth highlighting that, while early evidence supports a clinical benefit, their effect on the cardiac epigenome has not yet been proven as the underlying cardioprotective mechanism.

## Sodium-glucose cotransporter-2 inhibitors (SGLT2i's)

Two classes of metabolic therapies have been incidentally found to confer cardioprotective benefits, each originally developed to improve glycemic control for individuals with type 2 diabetes. Sodium-glucose cotransporter-2 inhibitors (SGLT2i's) were first identified as cardioprotective agents as a result of FDA-mandated cardiovascular outcomes trials (CVOTs), which found reductions in cardiovascular endpoints independent of their glucose-lowering effects (Anker et al, 2021; Fitchett et al, 2016; Fathi et al, 2021). Although inhibition of the SGLT2 transporter promotes a mild diuresis via the osmotic effects of glycosuria, this effect confers trivial hemodynamic benefits that does not account for their cardiovascular benefits among patients with HF, especially since

other diuretics have failed to demonstrate a similar benefit (Lopaschuk and Verma, 2020). The ventricular myocardium does not express SGLT2, prompting investigators to search for off target —and indirect—mechanisms of SGLT2i cardioprotection. Among these, in vitro studies demonstrated that the cardiomyocyte sodium-hydrogen exchanger (NHE-1) is inhibited by SGLT2i's, indirectly regulating intracellular calcium handling within isolated rabbit ventricular myocytes (Baartscheer et al, 2016).

SGLT2i's are known to exert epigenetic influence via class IIa and III HDACs. Dapagliflozin lowered *Camk2* expression in a rat model of isoproterenol-induced cardiomyopathy (Liu et al, 2023), which our group and others have shown to promote metabolic switching via nuclear exclusion of HDAC4 (Helmstadter et al, 2021; Lehmann et al, 2018). Another study used primary isolated rat fibroblast to show that treatment with either dapagliflozin or empagliflozin prevented the reduction of *Hdac6* expression caused by Angiotensin II and TGFβ exposure to lower collagen expression in cardiac fibroblasts (Ma et al, 2024). Similarly, SIRT1 is required for the cardioprotective effects of dapagliflozin in response to the transaortic constriction model of pressure overload-induced heart failure (Ren et al, 2022).

Regarding the upstream signaling that may confer these epigenetic alterations, a pre-print report found that SGLT2i activates by an off target mechanism pantothenate kinase 1 (PANK1), the rate-limiting enzyme that produces the CoA moiety from pantothenate, to stimulate metabolic flux in myocardial tissue from human subjects with heart failure (Forelli et al, 2024). Together, these data support that SGLT2 inhibition may promote epigenetic programming and/or myocardial metabolism.

## Incretin mimetic therapies (GLP-1RA's)

The second major cardiometabolic drug class identified through CVOTs comprises the family-wide effect of glucagon-like peptide-1 receptor agonists (GLP-1RAs), termed incretin mimetics, with landmark studies including LEADER (Liraglutide) (Marso et al, 2016b), SUSTAIN-6 (Semaglutide) (Marso et al, 2016a), REWIND (Dulaglutide) (Gerstein et al, 2019), and EXSCEL (Exenatide) (Holman et al, 2017). These agents confer significant cardiovascular protection, though their direct epigenetic effects on ventricular cardiomyocytes are unknown. GLP-1R expression within ventricular myocytes has not been consistently shown (Baggio et al, 2018; Taktaz et al, 2024), though emerging evidence suggesting that GLP-1RA effects extend beyond glycemic regulation. Genetic cardiomyocyte-specific disruption of *Glp1r* in mice failed to prevent cardioprotection by GLP-1RAs in the LAD ligation model of ischemic cardiomyopathy (Ussher et al, 2014). By contrast, in vitro GLP-1R agonism in non-cardiac tissue mitigates hyperglycemia-induced hypomethylation at the NF-κB and SOD2 promoters in human aortic endothelial cells, potentially encoding its anti-inflammatory and redox-stabilizing effects (Scisciola et al, 2023). Additionally, endopeptidase-generated GLP-1 metabolites promote PKA-mediated oxidative cytoprotection within the coronary endothelium and attenuate ischemic injury in vivo (Siraj et al, 2020). Glycemic response to GLP-1RA's is influenced by epigenetic variation, with hypomethylation of the *VTRNA2-1* promoter in peripheral blood correlating with significantly reduced therapeutic efficacy, particularly in individuals

carrying the rs2346018 allele (Lin et al, 2021). Therefore, while incretin mimetics deliver measurable cardiovascular benefits, further investigation is necessary to understand their longer-term impact on epigenetic regulation and metabolic remodeling in the heart.

## β-adrenergic receptor blockers and activators (β-ARB and β-ARA)

Among the oldest guideline-directed pharmacologic managements of heart failure are β-ARBs based on their established survival benefit (Packer et al, 1996; Bristow, 2011). And yet, despite clear clinical evidence supporting their implementation, the exact mechanism of β-ARB mediated cardioprotection remains incompletely understood. β adrenergic signaling in cardiomyocytes is well-characterized, but insights gained from adipocytes may offer additional perspectives. Catecholamine-induced lipolysis requires PKA and represents a well-established adaptive short-term mechanism to maintain body homeostasis and energy expenditure (Yang and Mottillo, 2020). We introduced the concept that intact lipolysis maintains cardiac function by counteracting the cardiotoxic production of anaplerotic metabolites of glycolysis (Backs et al, 2011; Jebessa et al, 2019; Lehmann et al, 2018). From this perspective, β-ARBs may potentiate the coupling of β-adrenergic receptors to PKA-dependent lipolysis (Zheng et al, 2005), whereby short-term β-adrenergic activation might paradoxically exert cardioprotection (Fig. 3). It is therefore fitting that selective β3-AR agonists, which also induce lipolysis in adipocytes, have entered clinical trials to improve diastolic performance in HFpEF (Pouleur et al, 2018; Balligand, 2013). Despite this mechanistic justification, clinical trials have not yet proven benefit of β3 agonism in cardiomyopathy. Among these negative trials, the SPHERE-HF was a randomized, double-blind, placebo-controlled phase II trial that measured the effects of β3 agonist mirabegron in patients with HFpEF, only to discover no difference in pulmonary vascular resistance (primary outcome), or right ventricular function (secondary outcome) (García-Álvarez et al, 2023). Thus, additional studies are needed to identify a translational application for the robust preclinical evidence supporting β3 agonism in cardiomyopathy. Further studies using conditional murine knockout alleles of β-adrenergic receptor subtypes (β1-, β2- and β3-adrenergic receptors) would help to distinguish the sub-type specific metabolic consequences of their inhibition and enable the development of more specific—and effective—therapeutic molecules. If proven correct, this perspective would challenge a prevailing viewpoint that β-ARBs confer cardioprotection via direct effect of catecholamines on calcium handling, and instead by exerting their effect on intermediary metabolic circuits.

## Targeting chromatin-modifying enzymes in cardiovascular and metabolic disease

The therapeutic targeting of chromatin-modifying enzymes is rapidly gaining traction in cardiovascular research, driven by growing recognition that their epigenetic and non-epigenetic mechanisms contribute to both the onset and progression of heart failure (Table 3). Regardless of the urgent need to identify and quantify the relative contributions of the direct versus indirect epigenetic targets within the metabo-epigenetic circuitries, the great potential of inhibiting these enzymes in metabolically driven heart disease must not be overlooked. Among these, histone deacetylase inhibition (HDACi) has shown promising cardioprotective effects in preclinical models, while ongoing development of bromodomain and extra-terminal domain (BET) and EZH2 inhibitors further expands the landscape of drug development pipelines, although translation of these therapies into the clinic remains challenged by issues of specificity, systemic toxicity, and unclear long-term efficacy in humans. Although lifestyle interventions such as exercise, weight loss, and dietary modification have been shown to influence the cardiac epigenome, these interventions engage a complex interplay of cellular signals that each warrant a dedicated in-depth discussion that falls beyond the scope of this review.

## HDAC inhibitors (HDACi's)

The direct targeting of epigenetic regulators, particularly via HDAC inhibition (HDACi), has become an exciting area of ongoing research and development (McKinsey, 2011; Lobera et al, 2013; Di Giorgio et al, 2014; Travers et al, 2021). In the myocardium, HDACs control the activity of many cardiac transcription factors, including GATA4, NFAT and MEF2, and have emerged as promising therapeutic targets in several heart diseases (Wright and Menick, 2016). The use of HDACi has already shown promising results on attenuating cardiac dysfunction and fibrosis (Wang et al, 2016).

Despite many efforts to advance precise control of enzymatic HDAC inhibition, panHDACi's (e.g., suberoylanilide hydroxamic acid, SAHA) have already been shown to demonstrate therapeutic benefits in preclinical hypertensive HFpEF and ischemic heart disease models (Wallner et al, 2020; Eaton et al, 2022; Travers et al, 2021; Jeong et al, 2018; Wang et al, 2016) but the underlying specific HDAC has not yet been clearly identified. As reviewed above class IIb HDAC inhibitors await clinical testing in HFpEF patients. PanHDACis also inhibit class IIa HDAC such as HDAC4 that also have been shown to specifically regulate cardiac metabolism via its non-enzymatic domain. At this point, we wish to emphasize the strong likelihood that the enzymatic and non-enzymatic actions of HDAC4 are closely interconnected in their biological function. Unpublished data from our group provide proof of concept of enzymatic class IIa HDAC inhibition as a potential new HFpEF therapy. Thus, it remains to be seen whether a subclass or isoform-specific HDAC inhibition approach might be translated to successful clinical trials. Targeted class IIa and class IIb HDAC inhibitors must first be optimized and then be tested in patients to obtain definitive answers for this question. An additional challenge will be the identification and stratification of patients that may derive the most benefit of class IIa and/or class IIb HDAC inhibitors.

However, the effects of HDACi—in particular class I HDACs—on non-cardiac tissues are widespread, as evidenced by their clinical usefulness as adjuvant anti-cancer therapies (Terranova-Barberio et al, 2017). Whether these concerns are justified for the inhibition of class IIa and class IIb HDACs remains to be tested in clinical trials. However—as reviewed above—class IIa HDACs in the myocardium have been shown to control the activity of many cardiac transcription factors, including GATA4, NFAT and MEF2, by a non-enzymatic mode of action, and have thus emerged as

**Table 3. Investigational epigenetic therapies.**

| Drug/compound | Epigenetic target | Mechanism of action | Clinical application | FDA status | Reference |
|---|---|---|---|---|---|
| Vorinistat (SAHA) | pan-HDACi (Class I/II) | ↓ LVH and fibrosis | Pressure overload HF | Preclinical (mouse) | (Eaton et al, 2022) |
| SK-7041 | | | | | (Kee et al, 2005) |
| Givinostat | | | Duchenne muscular dystrophy, Pressure overload HF | Phase IV (non-cardiac) | (Travers et al, 2021) |
| SRT2104 | SIRT1 | SIRT1 Agonist | Arterial stiffness, ventricular arrhythmias | Phase II | (Venkatasubramanian et al, 2016) |
| Resveratrol | SIRT1 (Class III) | Restores mitochondrial OXPHOS | Aging-related cardiac dysfunction | Preclinical (mouse) | (Sung et al, 2015) |
| TN-301 (TYA-018) | HDAC6 | ↓ HDAC6, ↓ LVH, fibrosis, and mitochondrial energy production | Cardiometabolic HFpEF | Phase Ib | (Ranjbarvaziri et al, 2024) |
| Sulforaphane | HDAC/DNMT | Activates Nrf2 antioxidant pathway | HFpEF, doxorubicin cardiomyopathy | Preclinical, Phase II (NCT05408559) | (Bose et al, 2018; Ma et al, 2018) |
| CDR132L | miR-132 | Inhibits miR-132, de-represses FoxO3, ↓ LVH and fibrosis | Chronic heart failure, Myocardial infarction | Phase II HF-REVERT (NCT05350969) | (Bauersachs et al, 2024) |
| AntimiR-21 | microRNA-21 | Inhibition of miR-21, | Myocardial Infarction, Ischemia-reperfusion injury | Preclinical (pig) | (Hinkel et al, 2020) |
| AntimiR-29 | microRNA-29 | Inhibition of miR-29, De-repression of Wnt signaling | Pathological cardiac remodeling and fibrosis | Preclinical (mouse) | (Sassi et al, 2017) |
| LNA-antimiR-208a | micoRNA-208a | Prevents pathological myosin switching and hypertrophy | HFrEF | Preclinical (rat) | (Montgomery et al, 2011) |
| JQ1 | BET proteins | Nonselective BET inhibition, ↓ inflammatory and profibrotic myocardial genes | HFrEF, Pulmonary arterial hypertension | Preclinical | (Duan et al, 2017) |
| GSK126 | EZH2 | Inhibits PRC2, ↓ CpG methylation | Myocardial infarction, HCM | Preclinical | (Aziz et al, 2023) |
| [a]Apabetalone (RVX-208) | BET proteins (BRD2/BRD3) | Selective bromodomain 2 inhibitor, ↓ inflammatory and profibrotic myocardial gene | MACE in type 2 diabetes after ACS | Phase III BETonMACE trial (NCT02586155) | (Ray et al, 2020) |
| [b]Decitabine [b]Azacitidine | DNMTs | Hypomethylate and suppress fibrotic and hypertrophic genes | Pressure overload HF | Preclinical (rat) | (Stenzig et al, 2018) |

The following table lists the drug or compound known to influence a chromatin-modifying enzyme with proposed applications in heart failure treatment.
HDACi histone deacetylase inhibitor, LVH left ventricular hypertrophy, HF heart failure, HFpEF heart failure with preserved ejection fraction, OXPHOS oxidative phosphorylation, HCM hypertrophic cardiomyopathy, PRC2 polycomb repressor complex 2.
[a]Phase III trial failed to demonstrate clinical benefit.
[b]Causes reversible cardiomyopathy.

promising therapeutic targets in heart disease (Kim et al, 2016; Wright and Menick, 2016; Lehmann et al, 2013). Augmenting the non-enzymatic functions of class IIa HDACs may allow more precise control than enzymatic inhibition.

## BET inhibitors

JQ1, a selective inhibitor of bromodomain and extra-terminal (BET) proteins such as BRD4, has demonstrated cardioprotective effects in preclinical models of heart disease. In murine models of pressure overload and myocardial infarction, JQ1 reduced cardiac hypertrophy, fibrosis, and inflammation while preserving ventricular function, primarily through repression of maladaptive gene networks including NF-κB and TGF-β signaling pathways (Duan et al, 2017). Single-cell transcriptomics have shown that JQ1 reprograms disease-associated cardiac fibroblasts and modulates endothelial and myeloid cell states, contributing to its broad anti-remodeling effects (Alexanian et al, 2021). Importantly, JQ1 spares physiological cardiac growth and attenuates hypertrophy in human iPSC-derived cardiomyocytes, suggesting disease-selective efficacy (Duan et al, 2017). However, clinical translation has proven difficult owing to its limited bioavailability and bioequivalent molecules failing to demonstrate clinical efficacy. Apabetalone (RVX-208) was a clinically tested BET inhibitor with selectivity for the second bromodomain (BD2), designed to reduce toxicity and off-target effects; however, this therapy failed to lower adverse cardiovascular events in diabetic patients enrolled in the BETon-MACE trial despite its proven pre-clinical anti-inflammatory and lipid-modifying properties (Ray et al, 2020). Therefore, ongoing investigations are underway to develop more potent BET inhibitors that still lack the concomitant toxicity and off-target consequences.

## EZH2 inhibitors

Owing to our recent work identifying EZH2 as a possible regulator of cardiac metabolism in patients with ischemic heart failure (Pepin et al, 2019b), the use of FDA-approved EZH2 inhibitors may provide a future benefit for patients with ischemic cardiomyopathy (Fig. 2). Evidence already supports a therapeutic role of small-molecule inhibition of EZH2 as a metabolic mechanism of protection against cellular de-differentiation and, consequently, malignant transformation of non-cardiac tissues (Shi et al, 2017; Duan et al, 2020). It has also been shown that suppressing EZH2 enhances the differentiation of cardiac fibroblasts into beating cardiomyocytes (Hirai and Kikyo, 2014). A preliminary report now supports the role of EZH2/PRC2 inhibitors to normalize lipid metabolism in ischemic cardiomyopathy using a murine model of myocardial infarction (Boel et al, 2023). Therefore, it remains plausible, yet unproven, that the induction of EZH2 moderates a metabolic switch within ischemic cardiomyocytes to produce a resting, non-contractile phenotype that can be reprogrammed for clinical HF recovery.

## Conclusion

The breakthroughs highlighted in this review underscore the interconnected roles of cardiac metabolism and histone-modifying enzymes in the pathogenesis of heart disease. Although

incompletely understood, we summarize the current evidence supporting the existence of "metabo-epigenetic circuits" that regulate the heart's response to both physiologic and pathologic metabolic perturbations. In the process, we feature yet-undefined elements of these circuits that merit further mechanistic exploration. It has become evident that histone-modifying enzymes influence metabolism directly and indirectly. The therapeutic targeting of this metabolic and/or epigenetic machinery, through either lifestyle or pharmacologic interventions, offers a promising arena for translational research. Several existing therapies for heart failure were originally developed for other metabolic indications and later found to confer cardioprotective benefits, implicating heart failure as itself a metabolic disease. Ultimately, leveraging the under-appreciated mechanisms of myocardial gene regulation is likely to enable precision-based "metabo-epigenetic" therapies to address the far-reaching, and seemingly intractable, consequences of heart disease and failure.

## Pending issues

- The fundamental architecture of metabo-epigenetic circuits in heart failure remains incomplete, requiring systematic mapping (i.e., with high-dimensional perturbation and multi-omic profiling) to define the missing nodes and metabolic signals that activate maladaptive gene programs.
- Key mechanistic paradoxes persist, including the divergent consequences of HDAC6 inhibition and the context-dependent effects of histone O-GlcNAcylation, necessitating inducible, dose-dependent, and cell type-specific interrogation studies.
- Translating metabo-epigenetic insights into therapy will require disease-relevant drug discovery platforms to develop selective epigenetic modulators, such as isoform-specific HDAC inhibitors and EZH2 blocking agents.

## Peer review information

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

## Acknowledgements

The authors would like to thank the reviewers for their thoughtful feedback and the editorial staff and at EMBO Molecular Medicine and the invitation to write this review. JB and AS are supported by grants from the Deutsche Forschungsgemeinschaft (Collaborative Research Center CRC1550 "Molecular Circuits of Heart Disease," INST 35/1699-1). Postdoctoral fellowship support for MEP was generously provided by the Humboldt-Forschungsstipendium through the Alexander von Humboldt Foundation. XG was supported by DZHK postdoc start-up funding.

## Author contributions

**Mark E Pepin**: Conceptualization; Formal analysis; Funding acquisition; Validation; Visualization; Writing—original draft; Writing—review and editing. **Xuemin Gong**: Conceptualization; Funding acquisition; Writing – original draft; Validation; Visualization; Writing—review and editing. **Almut Schulze**: Writing —review and editing. **Johannes Backs**: Conceptualization; Formal analysis; Funding acquisition; Validation; Visualization; Methodology; Writing—review and editing.

## Funding

## Disclosure and competing interests statement

JB holds a patent on "ABHD5 and partial HDAC4 fragments and variants as a therapeutic approach for the treatment of cardiovascular diseases." and is Founder of REVIER Therapeutics that develops first in class IIa HDAC inhibitors for the therapy of cardiometabolic disease. AS has no conflicts of interest to disclose. MEP serves as a clinical consultant for Suki AI.

