## [Peer Review File · EMBO Molecular Medicine]

Metabo-epigenetic circuits of heart failure: chromatin-modifying enzymes as determinants of metabolic plasticity

Mark Pepin, Xuemin Gong, Almut Schulze, and Johannes Backs

Corresponding author: Johannes Backs (Johannes.backs@cardioscience.uni-heidelberg.de)

Review Timeline:

Submission Date:	6th Aug 25
Editorial Decision:	1st Sep 25
Revision Received:	16th Sep 25
Editorial Decision:	26th Sep 25
Revision Received:	4th Oct 25
Accepted:	9th Oct 25

Editor: Lise Roth

Transaction Report:

1st Sep 2025

Dear Johannes,

Thank you for the submission of your review to EMBO Molecular Medicine. We have now received feedback from the experts who agreed to evaluate your manuscript (and had also reviewed your previous manuscript). As you will see from the reports below, they overall found the review well written, new, and interesting. They nevertheless make a few suggestions to improve the interest and impact of your work.

We would therefore welcome a revised version of your manuscript that would address these points. Please attach a covering letter giving details of the way in which you have handled each of the points raised by the referees.

- 1/ A .doc formatted version of the manuscript text (including Figure legends and tables).
- 2/ Separate figure files.
- 3/ A letter INCLUDING the reviewer's reports and your detailed responses to their comments.
- 4/ A glossary: EMBO Molecular Medicine articles are accompanied by a glossary explaining some of the terms used for laymen.
- 5/ Pending issues: At the end of each article, there is a box highlighting issues that still need further studies and where research efforts should converge.
- 6/ A 'disclosure statement and competing interests' statement (<https://www.embopress.org/competing-interests>).
- 7/ Up to 5 keywords.

For the figures, please note the following points:

- If there are certain aspects of your figure draft that are based upon assumptions or where the scientific data remains ambiguous, please add a comment so that we can work with you on an accurate depiction.
- If the figure or single panels of the figure have been adapted from a published figure, please add this information to the figure legend (e.g., 'Adapted from...' or 'Based on...').
- Please only re-use figures or parts of a figure if this is essential for understanding the concept communicated. If the figure contains re-used images or elements of images, please make sure that you have the permission/license to publish it (this also applies to your own previous work, if the journal you published in retains copyright). All re-used material must be explicitly cited.
- If you use an image data base for scientific iconography (e.g., BioRender), please let us know if you have a license that allows for publication in an academic journal.

Looking forward to receiving your revised manuscript at your earliest convenience,

With kind regards,

Lise

***** Reviewer's comments *****

Referee #1 (Remarks for Author):

I really enjoyed reading this review and I would give it to my students. Very well done and nicely balanced. I have nothing much to criticize, except for the usage of "epigenetic therapies", which is a bit far-fetched TMO. Unclear in the end what the precise MOA is and cause and consequence are difficult to untangle in this context, especially considering the fundamental impact on overall physiology.

Referee #2 (Remarks for Author):

This new version of the review written by Mark Pepin et al is interesting and clearer, focusing on the links between genetic modifications and metabolic alterations preceding the development of heart diseases.

However, I have a few comments.

1. A general comment concerns the link between genetic modifications affecting metabolic enzymes and the development of cardiovascular disease: could you discuss the fact that changing the substrate modifies the development of cardiovascular disease? Is this reversible? For example, if an obese person goes on a diet and loses weight, are the changes in the genome reversible? Is changing the substrate efficient ?
2. This also leads to reflection on therapies other than medication, such as diets (Mediterranean), exercise, etc. This is mentioned in Figure 3 but not in the text. Could you develop this aspect please ?
3. I would also qualify the conclusion and expand on this aspect, which could constitute complementary or alternative treatments to pharmacological treatments.
4. Figure 2 is not clear to me. It is too vague and confusing.

Referee #1

I really enjoyed reading this review and I would give it to my students. Very well done and nicely balanced. I have nothing much to criticize, except for the usage of "epigenetic therapies", which is a bit far-fetched TMO. Unclear in the end what the precise MOA is and cause and consequence are difficult to untangle in this context, especially considering the fundamental impact on overall physiology.

Response: We hope that our review offers an approachable, yet substantive, summary for learners and field experts alike. We appreciate the reviewer's contribution to achieving this aim. We do agree that the term "epigenetic therapies" is not precise enough and does not fully reflect what chromatin-modifying enzymes do. Thus, we took the opportunity to emphasize that future work needs to further dissect the non-epigenetic from the epigenetic effects of in particular HDACs, and we used the more neutral phrase "targeting chromatin-modifying enzymes" as we used this term also in the title. We have included a statement regarding the lack of definitive proof that therapies targeting epigenetic regulators confer cardioprotection via epigenetic mechanisms. This comment of the reviewer is well taken and timely. We strongly believe that in the next few years there will be more clarity on this topic.

Referee #2

This new version of the review written by Mark Pepin et al is interesting and clearer, focusing on the links between genetic modifications and metabolic alterations preceding the development of heart diseases. However, I have a few comments.

1. A general comment concerns the link between genetic modifications affecting metabolic enzymes and the development of cardiovascular disease: could you discuss the fact that changing the substrate modifies the development of cardiovascular disease? Is this reversible? For example, if an obese person goes on a diet and loses weight, are the changes in the genome reversible? Is changing the substrate efficient?

Author Response: The reviewer makes a valid point that small molecule inhibition of epigenetic modifiers has the potential to either confer off-target effects or involve non-epigenetic functions of the intended epigenetic regulator. In this regard, we now avoid the use of "epigenetic therapy" when referring to targeted disruption of chromatin-modifying enzymes given the lack of mechanistic certainty. We also now explicitly include a statement highlighting that, while targeting of epigenetic therapies shows early promise, the precise mechanism of cardioprotection has not been proven.

2. This also leads to reflection on therapies other than medication, such as diets (Mediterranean), exercise, etc. This is mentioned in Figure 3 but not in the text. Could you develop this aspect please?

Author Response: We appreciate the suggestion to include lifestyle factors in our description of potential interventions; we include a brief – yet admittedly inadequate – description of their effects on the cardiac epigenome. The complex cellular signals triggered by exercise, weight loss (e.g. via gastric surgery), and dietary modification as metabo-epigenetic interventions require dedicated sections. Therefore, although we retain a few studies from our original submission to highlight the known epigenetic effects of exercise (Lehmann et al. 2018), these lifestyle interventions engage a complex interplay of cellular signals such that each warrants its own dedicated in-depth review that falls beyond the scope of our review.

3. I would also qualify the conclusion and expand on this aspect, which could constitute complementary or alternative treatments to pharmacological treatments.

Author Response: We agree and have revised the conclusion to include lifestyle modification as a reasonable strategy to alter the metabo-epigenetic pathophysiology described, though again we reserve the in-depth discussion regarding their molecular benefits to future reviews. We thank the reviewer for requesting this addition.

4. Figure 2 is not clear to me. It is too vague and confusing.

Response: We have revised the legends of Figure 1 and 2 to better describe the mechanistic framework of metabolic regulatory gene network-based understanding of heart failure. The figure illustrates a rendition of the Proportional-Integral-Derivative (PID) control loop used by engineers to understand feedback loops used to control steady-state continuous processes.

Reference(s):

Lehmann, Lorenz H., Zegeye H. Jebessa, Michael M. Kreusser, Axel Horsch, Tao He, Mariya Kronlage, Matthias Dewenter, et al. 2018. "A Proteolytic Fragment of Histone Deacetylase 4 Protects the Heart from Failure by Regulating the Hexosamine Biosynthetic Pathway." *Nature Medicine* 24 (1): 62–72.

26th Sep 2025

Dear Prof. Backs, Dear Johannes,

Thank you for the submission of your revised manuscript to EMBO Molecular Medicine. Your manuscript is now ready to be accepted, once the following minor editorial issues are addressed:

- Please accept the previous changes and only keep in track changes mode any new modification in the text.
- "Journal Subject Terms" should be changed to "Keywords".
- Please merge "Funding Sources" with "Acknowledgements".
- Please rename "Conflict of Interest Disclosure" to "Disclosure and competing interests statement".
- Please add a "Pending Issues" section. We note that you incorporated a "Future Directions", which could be used to that effect. Pending issues should list a few issues that remain to be addressed. Please refer to previously published reviews for examples (the latest online: <https://www.embopress.org/doi/full/10.1038/s44321-025-00313-4>).
- We have sent your figures to our graphic designer, who will redraw them and get in touch with you once a first draft is ready (usually within 10 days).

Thank you for bearing with these last editorial matters.

Looking forward to receiving your revised manuscript,

With kind regards,

Lise

Lise Roth, Ph.D.
Senior Editor
EMBO Molecular Medicine

The authors addressed the remaining editorial issues.

9th Oct 2025

Dear Prof. Backs, Dear Johannes,

Thank you for submitting your revised manuscript. I am pleased to inform you that your manuscript is now accepted for publication and will be sent to our publisher once you approve the redrawn figures.

Your manuscript will be processed for publication by EMBO Press. It will be copy edited and you will receive page proofs prior to publication. Please note that you will be contacted by Springer Nature Author Services to complete licensing information.

This Review is free of charge. When you are contacted in a few weeks to sign your license agreement and review article proofs, please enter the following token into the relevant field in the Springer Nature author services system:[removed]

If you have any questions, please do not hesitate to contact the Editorial Office.
Thank you for your nice contribution to EMBO Molecular Medicine!

With kind regards,

Lise
